# QUANTIFYING AND MITIGATING THE IMPACT OF LABEL ERRORS ON MODEL DISPARITY METRICS

**Julius Adebayo**
Prescient Design / Genentech

**Melissa Hall**
Meta Inc.

**Bowen Yu**
Meta Inc.

**Bobbie Chern**
Meta Inc.

## ABSTRACT

Errors in labels obtained via human annotation adversely affect a model's performance. Existing approaches propose ways to mitigate the effect of label error on a model's downstream accuracy, yet little is known about its impact on a model's disparity metrics[1]. Here we study the effect of label error on a model's disparity metrics. We empirically characterize how varying levels of label error, in both training and test data, affect these disparity metrics. We find that group calibration and other metrics are sensitive to train-time and test-time label error—particularly for minority groups. This disparate effect persists even for models trained with noise-aware algorithms. To mitigate the impact of training-time label error, we present an approach to estimate the *influence* of a training input's label on a model's group disparity metric. We empirically assess the proposed approach on a variety of datasets and find significant improvement, compared to alternative approaches, in identifying training inputs that improve a model's disparity metric. We complement the approach with an automatic relabel-and-finetune scheme that produces updated models with, provably, improved group calibration error.

## 1 INTRODUCTION

Label error (noise) — mistakes associated with the label assigned to a data point — is a pervasive problem in machine learning (Northcutt et al., 2021). For example, 30 percent of a random 1000 samples from the Google Emotions dataset (Demszky et al., 2020) had label errors (Chen, 2022). Similarly, an analysis of the MS COCO dataset found that up to 37 percent (273,834 errors) of all annotations are erroneous (Murdoch, 2022). Yet, little is known about the effect of label error on a model's group-based disparity metrics like equal odds (Hardt et al., 2016), group calibration (Pleiss et al., 2017), and false positive rate (Barocas et al., 2019).

It is now common practice to conduct 'fairness' audits (see: (Buolamwini and Gebru, 2018; Raji and Buolamwini, 2019; Bakalar et al., 2021)) of a model's predictions to identify data subgroups where the model underperforms. Label error in the test data used to conduct a fairness audit renders the results unreliable. Similarly, label error in the training data, especially if the error is systematically more prevalent in certain groups, can lead to models that associate erroneous labels to such groups. The reliability of a fairness audit rests on the assumption that labels are *accurate*; yet, the sensitivity of a model's disparity metrics to label error is still poorly understood. Towards such end, we ask:

*what is the effect of label error on a model's disparity metric?*

We address the high-level question in a two-pronged manner via the following questions:

1. **Research Question 1**: What is the sensitivity of a model's disparity metric to label errors in training and test data? Does the effect of label error vary based on group size?

2. **Research Question 2**: How can a practitioner identify training points whose labels have the most *influence* on a model's group disparity metric?

---

[1]Group-based disparity metrics like subgroup calibration, false positive rate, false negative rate, equalized odds, and equal opportunity are more often known, colloquially, as *fairness metrics* in the literature. We use the term group-based disparity metrics in this work.

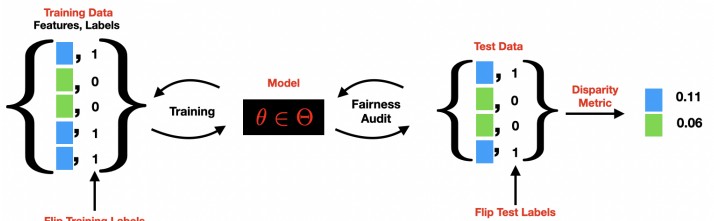

Figure 1: **A schematic of the test and train-time empirical sensitivity tests**. Here we show the model training and fairness audit pipeline. Our proposed sensitivity tests capture the effect of label error, in both stages, on the disparity metric. In the Test-time sensitivity test, we flip the label of a portion of the test data and then compare the corresponding disparity metric (group calibration for example) for the flipped dataset to the metrics for a standard model where the test labels were not flipped. In the Train-time sensitivity test, we flip the labels of a portion of the training set, and then measure the change in disparity metric to a standard model.

In addressing these questions, we make two broad contributions:

**Empirical Sensitivity Tests.** We assess the sensitivity of model disparity metrics to label errors with a label flipping experiment. First, we iteratively flip the labels of samples in the test set, for a fixed model, and then measure the corresponding change in the model disparity metric compared to an unflipped test set. Second, we fix the test set for the fairness audit but flip the labels of a proportion of the training samples. We then measure the change in the model disparity metrics for a model trained on the data with flipped labels. We perform these tests across a datasets and model combinations.

**Training Point Influence on Disparity Metric.** We propose an approach, based on a modification to the influence of a training example on a test example's loss, to identify training points whose labels have undue effects on any disparity metric of interest on the test set. We empirically assess the proposed approach on a variety of datasets and find a 10-40% improvement, compared to alternative approaches that focus solely on model's loss, in identifying training inputs that improve a model's disparity metric.

## 2 Setup & Background

In this section, we discuss notation, and set the stage for our contributions by discussing the disparity metrics that we focus on. We also provide an overview of the datasets and models used in the experimental portions of the paper.[2]

**Overview of Notation.** We consider prediction problems, i.e, settings where the task is to learn a mapping, $\theta : \mathcal{X} \times \mathcal{A} \rightarrow \mathcal{Y}$, where $\mathcal{X} \in \mathbb{R}^d$ is the feature space, $\mathcal{Y} \in \{0, 1\}$ is the output space, and $\mathcal{A}$ is a group identifier that partitions the population into disjoint sets e.g. race, gender. We can represent the tuple $(x_i, a_i, y_i)$ as $z_i$. Consequently, the $n$ training points can be written as: $\{z_i\}_{i=1}^n$. Throughout this work, we will only consider learning via empirical risk minimization (ERM), which corresponds to: $\hat{\theta} := \arg\min_{\theta \in \Theta} \frac{1}{n} \sum_i^n \ell(z_i, \theta)$. Similar to Koh and Liang (2017), we will assume that the ERM objective is twice-differentiable and strictly convex in the parameters. We focus on binary classification tasks, however, our analysis can be easily generalized.

**Disparity Metrics.** We define a group disparity metric to be a function, $\mathcal{GD}$, that gives a performance score given a model's probabilistic predictions ($\theta$ outputs the probability of belonging to the positive class) and 'ground-truth' labels. We consider the following metrics (We refer readers to the Appendix for a detailed overview of these metrics):

---

[2]We refer readers to the longer version of this work on the arxiv. Code to replicate our findings is available at: https://github.com/adebayoj/influencedisparity

| Dataset | Classes | $n$ | $d$ | Group | Source |
|---|---|---|---|---|---|
| CivilComments | 2 | $1,820,000$ | 768 | Sex | Koh and Liang (2017) |
| ACSIncome | 2 | $195,665$ | 10 | Sex, Race | Ding et al. (2021) |
| ACSEmployment | 2 | $378,817$ | 16 | Sex, Race | Ding et al. (2021) |
| ACSPublic Coverage | 2 | $138,554$ | 19 | Sex, Race | Ding et al. (2021) |
| Credit Dataset | 2 | $405,032$ | 6 | Sex | De Montjoye et al. (2015) |

Table 1: Overview of dataset characteristics for the datasets considered in this work.

1. **Calibration**: defined as $\mathbb{P}(\hat{y} = y | \hat{p} = p), \forall p \in [0, 1]$. In this work, we measure calibration with two different metrics: 1) Expected Calibration Error (ECE) (Naeini et al., 2015; Pleiss et al., 2017), and 2) the Brier Score (Rufibach, 2010) (BS).

2. (*Generalized*) **False Positive Rate (FPR)**: is $\mathcal{GD}_{\mathrm{fpr}}(\theta) = \mathbb{E}[\theta(x_i) \mid y_i = 0]$ (see Guo et al. (2017)),

3. (*Generalized*) **False Negative Rate (FNR)**: is $\mathcal{GD}_{\mathrm{fnr}}(\theta) = \mathbb{E}[(1 - \theta(x_i)) \mid y_i = 1]$,

4. **Error Rate (ER)**: is the $\mathcal{GD}_{\mathrm{er}}(\theta) = 1 - \mathrm{acc}(\theta)$.

We consider these metrics separately for each group as opposed to relative differences. For each dataset, we consider the protected data subgroup with the largest size as the majority group, and the group the smallest size is the minority group.

**Datasets.** We consider datasets across different modalities: 4 tabular, and a text dataset. A description of these datasets along with test accuracy is provided in Table 2. Each dataset contains annotations with a group label for both training and test data, so we are able to manipulate these labels for our empirical sensitivity tests. For the purposes of this work, we assume that the provided labels are the ground-truth—a strong assumption that nevertheless does not impact the interpretation of our findings.

**Model.** We consider three kinds of model classes in this work: 1) a logistic regression model, 2) a Gradient-boosted Tree (GBT) classifier for the tabular datasets, and 3) a ResNet-18 model. We only consider the logistic regression and GBT models for tabular data, while we fine-tune a ResNet-18 model on embeddings for the text data.

## 3    EMPIRICAL ASSESSMENT OF LABEL SENSITIVITY

In this section, we perform empirical sensitivity tests to quantify the impact of label error on test group disparity metrics. We conduct tests on data from two different stages of the ML pipeline: 1) Test-time (test dataset) and 2) Training-time (training data). We use as our primary experimental tool: label flipping, i.e., we flip the labels of a percentage of the samples, uniformly at random in either the test or training set, and then measure the concomitant change in the model disparity metric. We assume that each dataset's labels are the ground truth and that flipping the label results in label error for the samples whose labels have been overturned. Recent literature has termed this setting synthetic noise, i.e., the label flipping simulates noise that might not be representative of real-world noise in labels (Arpit et al., 2017; Zhang et al., 2021; Jiang et al., 2020).

### 3.1    SENSITIVITY TO TEST-TIME LABEL ERROR

**Overview & Experimental Setup.** The goal of the test-time empirical test is to measure the impact of label error on the group calibration error of a fixed model. Consider the setting where a model has been trained, and a fairness assessment is to be conducted on the model. What impact does label error, in the test set used to conduct the audit, have on the calibration error on the test data? The test-time empirical tests answer this question. Given a fixed model, we iteratively flip a percentage of the labels, uniformly at random, ranging from zero to 30 percent in the test data. We then estimate the model's calibration using the modified dataset. Critically, we keep the model fixed while performing these tests across each dataset.

**Results.** In Figure 2, we report results of the label flipping experiments across 6 tasks. On the horizontal axis, we have the percentage of labels flipped in the test dataset, while on the vertical

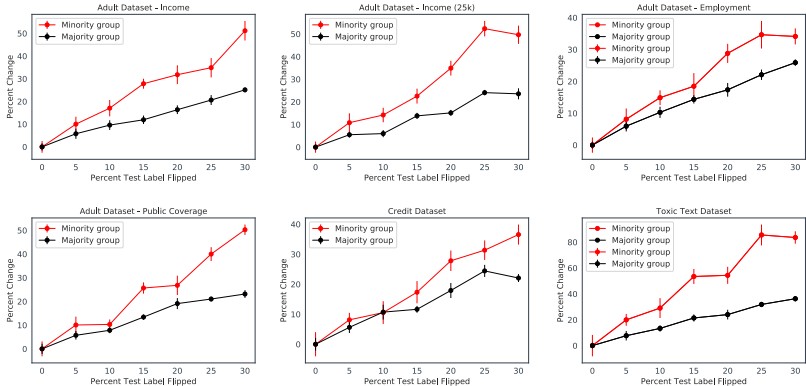

Figure 2: **Test-time Label Flipping Results across**. For each dataset, we plot the percent change in calibration error versus the corresponding percentage change in label error. Here, we plot the minority (smallest) group as well as the majority (largest) group. These two groups represent two ends of the spectrum for the impact of label error. We observe that across all datasets, the minority group incurs higher percentage change in group calibration compared to the majority group.

axis, we have the percentage change in the model's calibration. For each dataset, we compute model calibration for two demographic groups in the dataset, the majority and the minority—in size–groups. We do this since these two groups constitute the two ends of the spectrum in the dataset. As shown, we observe a more distinctive effect for the minority group across all datasets. This is to be expected since flipping even a small number samples in the minority group can have a dramatic effect on test and training accuracy within this group. For both groups, we observe a changes to the calibration error. For example, for the Income prediction task on the Adult dataset, a 10 percent label error induces at least a 20 percent change in the model's test calibration error. These results suggest that test-time label error has more pronounced effects for minority groups. Similarly, we observe for other disparity metrics (See Appendix) across all model classes that increases in percentage of labels flipped disproportionately affects the minority group.

## 3.2 Sensitivity to Training Label Error

**Overview & Experimental Setup.** The goal of the training-time empirical tests is to measure the impact of label error on a trained model. More specifically, given a training set in which a fraction of the samples' labels have been flipped, what effect does the label error have on the calibration error compared to a model trained on data without label error? We simulate this setting by creating multiple copies of each of the datasets where a percentage of the training labels have been flipped uniformly at random. We then assess the model calibration of these different model using the same fixed test dataset. Under similar experimental training conditions for these models, we are then able to quantify the effect of training label error on a model's test calibration error. We conduct this analysis across all dataset-model task pairs.

**Results & Implications.** We show the results of the training-time experiments in Figure 3. Similar to the test-time experiments, we find minority groups are more sensitive to label error than larger groups. Specifically, we find that even a 5 percent label error can induce significant changes in the disparity metrics, of a model trained on such data, for these groups.

A conjecture for the higher sensitivity to extreme training-time error is that a model trained on significant label error might have a more difficult time learning patterns in the minority class where there are not enough samples to begin with. Consequently, the generalization performance of this model worsens for inputs that belong to the minority group. Alternatively, in the majority group, the proportion of corrupted labels due to label error is smaller. This might mean that uniform flipping does not affect the proportion of true labels compared to the minority group. Even though the majority group exhibits label error, there still exists enough samples with true labels such that a model can learn the underlying signal for the majority class.

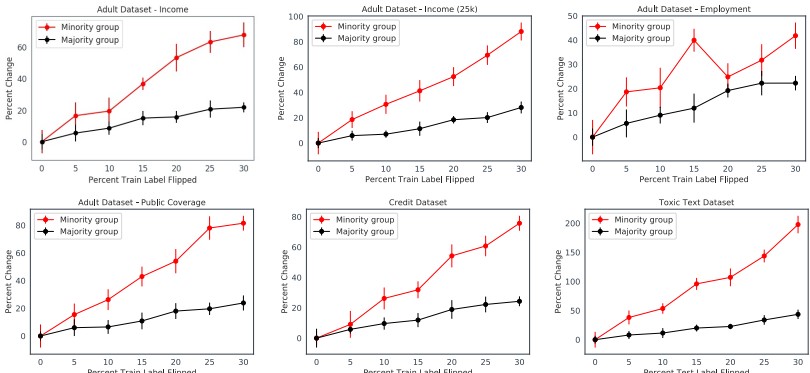

Figure 3: **Training-time Label Flipping Results**. For each dataset, we plot the percent change in calibration error versus the corresponding percentage change in label error for the training set. Here, we plot the minority (smallest) group as well as the majority (largest) groups by size. Similar to the test-time setting, we observe that across all datasets, the minority group incurs higher percentage change in group calibration compared to the majority group. However, we observe a larger magnitude change for the minority groups.

A second important finding is that overparameterization seems to confer more resilience to training label error. We find that for the same levels of training label error, an overparametrized model is less sensitive to such change compared to a model with a smaller number of parameters. Recent work suggests that models that learn functions that are more aligned with the underlying target function of the data generation process are more resilient to training label error (Li et al., 2021). It might be that compared to linear and tree-based models, an overparametrized deep net is more capable of learning an aligned function.

## 3.3  NOISE-AWARE ROBUST LEARNING HAS DISPARATE IMPACT

**Overview & Experimental Setup.** We now assess whether training models with noise-aware algorithmic interventions (e.g. robust loss functions (Ma et al., 2020; Ghosh et al., 2017)) results in models whose disparity metrics have reduced sensitivity to label error in the training set. We test this hypothesis on a modified Cifar-10 dataset following the setting of Hall et al. (2022). Specifically, the Cifar-10 dataset is modified to a binary classification setting along with group labels by inverting a subset of each class's examples. Given a specified parameter $\epsilon \in [0, 1/2]$, a $\frac{1}{2} - \epsilon$ of the negative class is inverted, while a $\frac{1}{2} + \epsilon$ of the positive class is inverted leading to $2\epsilon$ fraction of one group of samples and $1 - 2\epsilon$ of the other group. In all experiments we set $\epsilon = 0.15$ for a 30 percent minority group membership. We replicate the label flipping experiment on the task

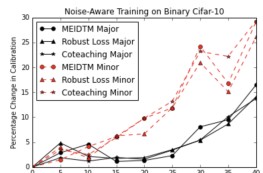

Figure 4: **Effect of Noise-aware algorithms on group calibration**.

with a Resnet-18 model. We test the MEIDTM (Cheng et al., 2022), DivideMix (Li et al., 2020), and a robust loss approach (Ghosh et al., 2017).

**Results.** At a high level, for the majority group, we find that group calibration remains resilient to low rates of label error (below 25 percent). At high rates (>30 percent label error), we start to see increased sensitivity. However, for the minority group (30 percent of the dataset), we observe group calibration remains sensitive to label error even at low levels. This finding suggests that noise-aware methods show are more effective for larger groups in the data. A similar observation has also been made for other algorithmic interventions like Pruning (Tran et al., 2022; Hooker et al., 2019), differential privacy (Bagdasaryan et al., 2019), selective classification (Jones et al., 2020) and adversarial training (Xu et al., 2021).

## 4 INFLUENCE OF TRAINING LABEL ON TEST DISPARITY METRIC

We now present an approach for estimating the 'influence' of perturbing a training point's label on a disparity metric of interest. We consider: 1) up-weighting a training point, and 2) perturbing the training label.

**Upweighting a training point.** Let $\hat{\theta}_{-z_i}$ be the ERM solution when a model is trained on all data points, $\{z_j\}_{j=1}^n$, except $z_i$. The influence, $\mathcal{I}_{\text{up,params}}$, of datapoint, $z_i$, on the model parameters is then defined: $\hat{\theta}_{-z_i} - \hat{\theta}$. This measure indicates how much the parameters change when the model is 'refit' on all training data points except $z_i$. Koh and Liang (2017) give a closed-form estimate of this quantity as:

$$\mathcal{I}_{\text{up,params}} \overset{\text{def}}{=} \left. \frac{d\hat{\theta}_{\epsilon, z_i}}{d\epsilon} \right|_{\epsilon=0} = -H_{\hat{\theta}}^{-1} \nabla_\theta \ell(z_i, \hat{\theta}), \tag{1}$$

where $H$ is the hessian, i.e., $H_{\hat{\theta}} \overset{\text{def}}{=} \frac{1}{n} \sum_{i=1}^n \nabla_\theta^2 \ell(z_i, \theta)$.

The loss on a test example, $\ell(z_t, \hat{\theta})$, is a function of the model parameters, so using the chain-rule, we can estimate the influence, $\mathcal{I}_{\text{up,loss}}(z_i, z_t)$, of a training point, $z_i$, on $\ell(z_t, \hat{\theta})$ as:

$$\mathcal{I}_{\text{up,loss}}(z_i, z_t) \overset{\text{def}}{=} \left. \frac{d\ell(z_t, \hat{\theta}_{\epsilon, z_i})}{d\epsilon} \right|_{\epsilon=0} = -\nabla_\theta \ell(z_t, \hat{\theta})^\top H_{\hat{\theta}}^{-1} \nabla_\theta \ell(z_i, \hat{\theta}). \tag{2}$$

**Perturbing a training point's label.** A second notion of influence that Koh and Liang (2017) study is how perturbing a training point leads to changes in the model parameters. Specifically, given a training input, $z_i$, that is a tuple $(x_i, y_i)$, how would the perturbation, $z_i \rightarrow z_{i,\delta}$, which is defined as $(x_i, y_i) \rightarrow (x_i, y_i + \delta)$, change the model's predictions? Koh and Liang (2017) give a closed-form estimate of this quantity as:

$$\mathcal{I}_{\text{pert,loss,y}}(z_j, z_t) \approx -\nabla_\theta \ell(z_t, \hat{\theta}_{z_{j,\delta}, -z_j})^\top H_{\hat{\theta}}^{-1} \nabla_y \nabla_\theta \ell(z_j, \hat{\theta}). \tag{3}$$

**Adapting influence functions to group disparity metrics.** We now propose modifications that allow us to compute the influence of a training point on a test group disparity metric (See Appendix D for longer discussion). Let $S_t$ be a set of test examples. We can then denote $\mathcal{GD}(S_t, \hat{\theta})$ as the group disparity metric of interest, e.g., the estimated ECE for the set $S_t$ given parameter setting $\hat{\theta}$.

**Influence of upweighting a training point on a test group disparity metric.** A group disparity metric on the test set is a function of the model parameters; consequently, we can apply the chain rule to $\mathcal{I}_{\text{up,params}}$ (from Equation 1) to estimate the influence, $\mathcal{I}_{\text{up,disparity}}$, of up-weighting a training point on the disparity metric as follows:

$$\mathcal{I}_{\text{up,disparity}}(z_i, S_t) \overset{\text{def}}{=} \left. \frac{d\mathcal{GD}(S_t, \hat{\theta}_{\epsilon, z_i})}{d\epsilon} \right|_{\epsilon=0} = -\nabla_\theta \mathcal{GD}(S_t, \hat{\theta})^\top \left. \frac{d\hat{\theta}_{\epsilon, z_i}}{d\epsilon} \right|_{\epsilon=0},$$
$$= -\nabla_\theta \mathcal{GD}(S_t, \hat{\theta})^\top H_{\hat{\theta}}^{-1} \nabla_\theta \ell(z_i, \hat{\theta}). \tag{4}$$

We now have a closed-form expression for a training point's influence on a test group disparity metric.

**Influence of perturbing a training point's label on a test group disparity metric.** We now consider the influence of a training label perturbation on a group disparity metric of interest. To do this, we simply consider the group disparity metric function as the quantity of interest instead of the test loss. Consequently, the closed-form expression for the influence of a modification to the training label on disparity for a given test set is:

$$\mathcal{I}_{\text{pert,disparity,y}}(z_j, S_t) \approx -\nabla_\theta \mathcal{GD}(S_t, \hat{\theta})^\top H_{\hat{\theta}}^{-1} \nabla_y \nabla_\theta \ell(z_j, \hat{\theta}). \tag{5}$$

With Equations 4 and 5, we have the key quantities of interest that allows us to rank training points, in terms of influence, on the test group disparity metric.

# 5 IDENTIFYING AND CORRECTING TRAINING LABEL ERROR

In this section, we empirically assess the modified influence expressions for calibration across these datasets for prioritizing mislabelled samples. We find that the prioritization scheme shows improvement, compared to alternative approaches. In addition, we propose an approach to automatically correct the labels identified by our proposed approach.

## 5.1 IDENTIFYING LABEL ERROR

**Overview & Experimental Question.** We are interested in surfacing training points whose change in label will induce a concomitant change in a test disparity metric like group calibration. Specifically, we ask: When the training points are ranked by influence on test calibration, are the most highly influential training points most likely to have the wrong labels? We conduct our experiments to directly measure a method's ability to answer this question.

**Experimental Setup.** For each dataset, we randomly flip the labels of $10 - 30$ percent of the training samples. We then train on this modified dataset. In this task, we have direct access to the ground-truth of the exact samples whose labels were flipped. This allows us to directly compare the performance of our proposed methods to each of the baselines on this task. We then rank training points using a number of baseline approaches as well as the modified influence approaches. For the top $50$ examples, we consider what fraction of these examples had flipped labels in the training set. We discuss additional experimental details in the Appendix.

**Approaches & Baselines.** We consider the following methods: 1) **IF-Calib**: The closed-form approximation to the influence of a training point on the test calibration; 2) **IF-Calib-Label**: The closed-form approximation to the influence of a training point's label on the test calibration; 3) **Loss**: A baseline method which is the training loss evaluated at each data point in the training set. The intuition is that, presumably, more difficult training samples will have higher training loss. We also consider several additional baselines that we discuss in the Appendix.

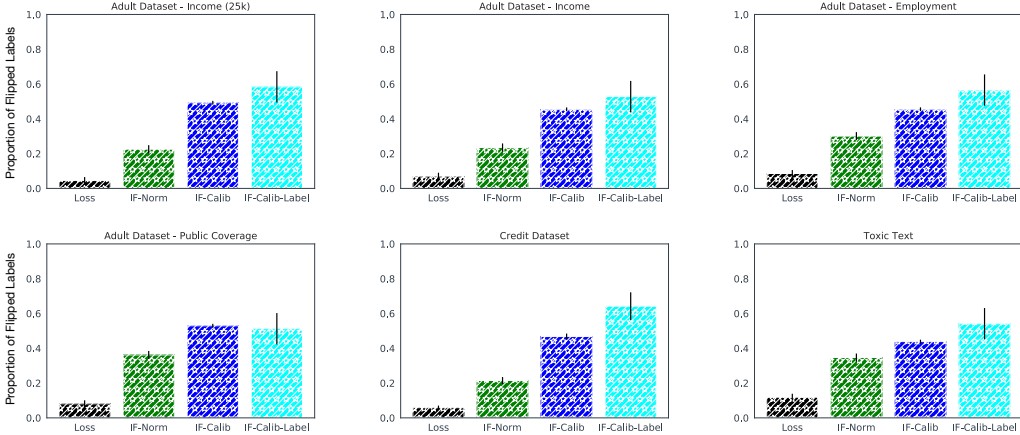

Figure 5: **Empirical Results for Training Point Ranking Across 6 datasets**. For the top 50 most influential examples, we show the proportion of samples whose labels were flipped in the training data.

**Results: Prioritizing Samples.** In Figure 5, we show the performance of the two approximations that we consider in this work as well as two baselines. We plot the fraction of inputs, out of the top ranked $50$ ranked training points, whose labels were flipped in the training set. The higher this proportion, then the more effective an approach is in identifying the samples that likely have wrong labels. In practice, the goal is to surface these training samples and have a domain expert inspect them. If a larger proportion of the items to be inspected are mislabeled, then a higher proportion of training set mistakes, i.e. label error, can be fixed. Across the different datasets, we find a 10-40 percent improvement, compared to baseline approaches, in identifying critical training data points whose labels need to be reexamined.

We find the loss baseline to be ineffective for ranking in our experiments. A possible reason is that modern machine learning models can typically be trained to 'memorize' the training data; resulting in settings where a model has low loss even on outliers or mislabeled examples. In such a case, ranking by training loss for a sample is an ineffective ranking strategy. We find that the noise-aware approaches perform similarly to the IF-Norm baseline. We defer the results of the uncertainty-based baselines and the noise-aware methods to Appendix (Section F). We find that these baselines also underperform our proposed approaches.

## 5.2   CORRECTING LABEL ERROR

We take label error identification one step further to automatically relabelling inputs that have identified as critical. We restrict our focus to binary classification where the label set is $\{0, 1\}$, and the corresponding relabelling function is simply $1 - y_i$, where $y_i$ is the predicted label.

**Setup & Experiment**: We consider the logistic regression model across all tasks for a setting with 20 percent training label error. We consider calibration as the disparity function of interest. We then rank the top 20 percent of training points by label-disparity influence, our proposed approach. For these points, we apply the relabelling function, and then fine-tune the model for an additional epoch with the modified labels.

**Results:** First, we observe an improvement, in group calibration, across all groups, with larger improvement coming from the smallest group. As expected, we also observe a decrease in the average loss for the overall training set. These results point to increasing promise of automatic relabeling.

**Theoretical Justification.** We now present a theorem that suggests that the influence priorization and relabeling scheme described above provably leads to better calibrated models.

**Theorem 1.** *Given a $\kappa$-strongly convex loss function $\ell(.,.)$, with $\kappa > 0$, a training dataset, $\mathcal{D}$, where $A$ indexes the data groups, and a model, $\hat{\theta} : x_i \to y_i$, optimized via $\ell(.,.)$ that maps inputs to labels. Let $\mathcal{Q}$ be a set of test examples all belonging to group $A = a$, where $\mathrm{ECal}_{\mathcal{Q}}(\hat{\theta})$ is the expected calibration error of $\hat{\theta}$ on the set $\mathcal{Q}$. In addition, let $\mathcal{D}_{A=a}$ be the set of problematic training examples, belonging to group $a$, prioritized based on influence, i.e., $\mathcal{I}_{\mathrm{pert,calib},y^i}(x_a^i, \mathcal{Q}) > 0$. We term a model trained on a different training set ($\mathcal{D}_+$) where the problematic examples have been relabeled to be $\hat{\theta}_R$. Analogously, the expected calibration error of this new model on the set $\mathcal{Q}$ is $\mathrm{ECal}_{\mathcal{Q}}(\hat{\theta}_R)$. We have that:*

$$\mathrm{ECal}_{\mathcal{Q}}(\hat{\theta}_R) \leq \mathrm{ECal}_{\mathcal{Q}}(\hat{\theta}).$$

We defer the proof to the Appendix. Theorem 1 suggests that when a model is trained on a relabeled dataset, following the influence prioritization scheme, the expected group calibration of the retrained model should be lower than that of a model trained on a dataset that has not been relabeled.

## 6   RELATED WORK

We discuss directly related work here, and defer a longer discussion to Section A of the Appendix.

**Impact of Label Error/Noise on Model Accuracy.** Learning under label error falls under the category more commonly known as *learning under noise* (Frénay and Verleysen, 2013; Natarajan et al., 2013; Bootkrajang and Kabán, 2012). *Noise* in learning can come from different either input features or the labels. In this work, we focus on label error—categorization mistakes associated with the label in both the test and training data. Previous work focused primarily on the effect of label error in the training data; however, we advance this line of work to investigate the effect of label error in the test data used to conduct a fairness audit on the reliability of the audit. Model resilience to training label error has been studied for both synthetic (Arpit et al., 2017; Zhang et al., 2021; Rolnick et al., 2017) and real-world noise settings (Jiang et al., 2020). A major line of inquiry is the development of algorithmic approaches to learn accurate models given a training set with noisy labels. These approaches include model regularization (Srivastava et al., 2014; Zhang et al., 2017), bootstrap (Reed et al., 2014), knowledge distillation (Jiang et al., 2020), instance weighting (Ren et al., 2018; Jiang and Nachum, 2020), robust loss functions (Ma et al., 2020; Ghosh et al., 2017), or trusted data (Hendrycks et al., 2018), joint training (Wei et al., 2020), mixture models in semi-supervised learning (Li et al.,

2020), and methods to learn a transition matrix that captures noise dependencies (Cheng et al., 2022). In contrast to this line of work, we primarily seek to identify the problematic instances that need to be relabelled, often by a human labeler, and not automatically learn a model that is robust to label error.

**Impact of Label Error on Model *'Fairness'*.** This work contributes to the burgeoning area that studies the impact of label error on a model's 'fairness' (termed 'group-based disparity' in this paper) metrics. Fogliato et al. (2020) studied a setting in which the labels used for model training are a noisy proxy for the true label of interest, e.g., predicting rearrest as a proxy for rearrest. Wang et al. (2021) considers an ERM problem subject to group disparity constraints with group-dependent label noise, and provides theoretical results along with a scheme to obtain classifiers that are robust to noise. Different from their setting, we consider unconstrained ERM (no fairness constraints during learning). Similarly, Konstantinov and Lampert (2021) study the effect of adversarial data corruptions on fair learning in a PAC model. Jiang and Nachum (2020) propose a re-weighting scheme that is able to correct for label noise.

**Influence Functions & Their Uses.** Influence functions originate from robust statistics where it is used as a tool to identify outliers (Cook and Weisberg, 1982; Cook, 1986; Hampel, 1974). Koh and Liang (2017) introduced influence functions for modern machine learning models, and used them for various model debugging tasks. Most similar to our work, Sattigeri et al. (2022) and Li and Liu (2022) also consider the influence of a training point on model's disparity metric, and present intriguing results that demonstrate that reweighting training samples can improve a model's disparity metrics. Here, we focus specifically on the role of mislabeled examples; however, our goal aligns with theirs. Similarly, Kong et al. (2021) propose RDIA, a relabelling scheme based on the influence function that is able to provably correct for label error in the training data. RDIA identifies training samples that have a high influence on the test loss for a validation set; however, we focus on identifying training samples that influence a group-disparity metric on a test/audit set. We also rely on their technical results to prove Theorem 1.

In recent work, De-Arteaga et al. (2021) study expert consistency in data labeling and use influence functions to estimate the impact of labelers on a model's predictions. Along similar direction, Brunet et al. (2019) adapt the influence function approach to measure how removing a small part of a training corpus, in a word embedding task, affects test bias as measured by the word embedding association test Caliskan et al. (2017). Feldman and Zhang (2020) use influence functions to estimate how likely a training point is to have been memorized by a model. More generally, influence functions are gaining widespread use as a tool for debugging model predictions (Barshan et al., 2020; Han et al., 2020; Yeh et al., 2018; Pruthi et al., 2020). Different from these uses of influence functions, here we isolate the effect of a training point's label on a model's disparity metric on a audit data.

# 7 CONCLUSION

In this paper, we sought to address two key questions: *1) What is the impact of label error on a model's group disparity metric, especially for smaller groups in the data;* and *2) How can a practitioner identify training samples whose labels would also lead to a significant change in the test disparity metric of interest?* We find that disparity metrics are, indeed, sensitive to test and training time label error particularly for minority groups in the data. In addition, we present an approach for estimating the 'influence' of perturbing a training point's label on a disparity metric of interest, and find a 10-40% improvement, compared to alternative approaches, in identifying training inputs that improve a model's disparity metric. We present an approach to estimate the effect of a training input's label on a model's group disparity metric. Lastly, perform a simple automatic relabel-and-finetune scheme that produces updated models with, provably, improved group calibration error.

Our findings come with certain limitations. In this work, we focused on the influence of label error on disparity metrics. However, other components of the ML pipeline can also impact downstream model performance. The proposed empirical tests simulate the impact of label error; however, it might be the case that real-world label error is less pernicious to model learning dynamics than the synthetic flipping results suggest. Ultimately, we see our work as helping to provide insight and as an additional tool for practitioners seeking to address the challenge of label error particularly in relation to a disparity metric of interest.

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
