# OpenReview forum: "Quantifying and Mitigating the Impact of Label Errors on Model Disparity Metrics"
_ICLR.cc/2023/Conference — ICLR 2023 poster_

### Official Review · Reviewer_DK6U · 2022-10-24

**Confidence:** 3
**Clarity, Quality, Novelty And Reproducibility:** 1. Their theoretical analysis seems t…
**Correctness:** 4
**Technical Novelty And Significance:** 3
**Empirical Novelty And Significance:** 3
**Recommendation:** 8

**Strength And Weaknesses:**

Strength

1. They try to answer two significant questions regarding label errors and fairness:
    What is the sensitivity of a model’s disparity metric to label errors in training and test data? Does the effect of label error vary based on group size?
    How can a practitioner identify training points whose labels have the most influence on a model’s group disparity metric?
2. Various disparity metrics, models, and datasets are considered
    model disparity metrics: expected calibration error (ECE), the Brier Score (BS), False Positive Rate (FPR), False Negative Rate (FNR), and Error Rate (ER).
    datasets across different modalities: 4 tabular, and a text dataset.
    models: a logistic regression model, a Gradient-boosted Tree (GBT), and ResNet-18.
3. The closed form influence function is potentially useful for regularizing or mitigating the label error while training.

Weaknesses
1. Although multiple datasets are used, how to make sure that the conclusions and analysis drawn is not dataset-specific is still an open question.
2. The label flipping is indeed useful for empirical results but not interesting as a technical contribution.


**Summary Of The Paper:**

1. This paper studies the effect of label error on a model’s group-based disparity metrics. Differences in terms of calibration error can be observed for the minority (smallest) group and the majority (largest) group.
2. They also propose an approach (influence function) to estimate how changing a single training input’s label affects a model’s group disparity metric on a test set. Based on the influence function, they can identify training points that have a high effect on a model’s test disparity metric.


**Summary Of The Review:**

1. The overall quality of the paper is good, and the key ideas are clear enough to make reviewers easy to follow.
2. If as they claimed, label errors’ influence on group disparity is not yet covered by literature, this is a strong submission.

---

> ### Author Response · Authors · 2022-11-17
> **Response to reviewer DK6U**
>
> We thank you for your comments and feedback. In addition to the general updates, we address your concerns here.
> ___
>
> **Reliability of Empirical Conclusions**\
> As we note in the general comment, our results and findings hold even as we expand to several other datasets. The consistency of the results across these datasets suggests that our conclusion is not dataset dependent. We have now added additional datasets and suggestive theoretical insights that indicate that our findings are not an artifact due to a specific experimental or dataset setting.
>
> **Label Flipping:** We agree with you about label flipping. We do not claim it as a technical contribution in this paper. As a matter of fact, label flipping has been used across several other settings. Following previous work, we simply use label flipping as an experimental tool to test the effect of label error on a model's disparity metric.
>
>
> >Their theoretical analysis seems to be more interesting. But is the chain rule analysis covered by other work? This is one part which I am not sure about regarding novelty.
>
> We have updated the paper and the related work to more clearly delineate our contributions. As we note in the general comment, as far as we are aware, our collection of contributions towards understanding the effect of label error on a model's disparity metrics reveal new insights not previously present in the prior literature.
>
> On the use of influence functions: influence functions have been previously used in the fairness setting as we discuss in the paper, but not in the specific way that we propose here. First, we are interested in the effect of a change in the label on a model's disparity metric. Others have considered how a change in the entire sample affects the model's prediction, or validation accuracy. Here, we are interested in the isolated effect of the label on a fairness property which has not been previously considered. Again, we reiterate that our collection of contributions goes beyond the proposed approach.
>
> **Label Noise vs Label Error**\
> In this paper, label error refers to an error in the specific label of a single or a group of samples in a dataset. For example, a digit 1 in the MNIST dataset whose training label is indicated as a 8 exhibits label error. Specifically, we use the term to target classification tasks. More generally in the literature, label noise also captures regressions settings where the target variable is a scalar. Label noise subsumes label error as used in this paper.
>
> We cited the "Fair Classification with Group-Dependent Label Noise" paper in our original version. The setting of the paper also maps to our definition of label error. More specifically, the paper considers an ERM problem subject to fairness constraints like equality of true positive rates where the training data has been corrupted with group (protected attribute) dependent noise. The paper provides theoretical results along with a scheme to obtain classifiers, in such settings, that are still robust to noise. Different from their setting, we consider unconstrained ERM (no fairness constraints during learning), and quantify the sensitivity of the resulting classifiers to both uniform and group-dependent noise. Even though we only consider classification tasks in this paper, the proposed approach can be applied to regression settings as well; however, we leave this for future work. We have updated the related work section with this discussion.
>
> Thank you for the feedback, we hope we have adequately addressed your concerns. We will be happy to answer any additional questions. We encourage you to reconsider their score in light of our updates.

---

> > ### Author Response · Authors · 2022-12-01
> > **Happy to provide additional clarification**
> >
> > We hope our response clarified your initial concerns/questions. We would be happy to provide further clarifications where necessary.

---

> > > ### Author Response · Authors · 2022-12-06
> > > **Have your previous concerns been addressed?**
> > >
> > > We are reaching out to check whether the reviewer has any additional questions based on our previous response.

---

### Official Review · Reviewer_oCK7 · 2022-10-24

**Confidence:** 3
**Correctness:** 3
**Technical Novelty And Significance:** 3
**Empirical Novelty And Significance:** 3
**Recommendation:** 6

**Clarity, Quality, Novelty And Reproducibility:**

The paper is clearly written in general, and ample details are provided to help reproduce the results shown in the paper.

**Strength And Weaknesses:**

Strength:
- This paper is very well organized and written in general. Most of the claims are supported by ample experimental analysis.
- The problem of concern has a unique fairness perspective, which has great practical significance.

Weaknesses:
- Learning with noisy labels is a widely studied topic, especially in the context of neural networks. As an empirical paper, it would be nice if the authors of the paper can conduct some additional analysis to show the effect of some of the recently proposed noise-robust algorithms on such group-based disparity metrics.
- Similar to the above point, it would be nice if the authors of the paper benchmarked the proposed "IF-Calib-Label" against some other recently proposed noise-robust algorithms that can potentially identify label errors.

**Summary Of The Paper:**

This paper considers an important problem of label noise in the training data. Specifically, it studies the effect of label error on a model's group-based disparity metrics, with more focus on smaller groups in the data. Then, the authors of the paper take a step further by considering a method based on influence function to identify training samples that significantly impact the test disparity metric of interest. The authors of the paper conduct a series of experiments to answer these questions and offer valuable insight into this important problem.

**Summary Of The Review:**

Despite the interesting perspective and a well series of well-conducted experiments, I feel like the authors of the paper can provide some additional experimental insight to the paper, as suggested above. As such, I recommend a weak accept for now.

---

> ### Author Response · Authors · 2022-11-17
> **Response to reviewer oCK7**
>
> We thank you for your comments and feedback. In addition to the general updates, we address your concerns here.
> ___
>
> **Comparison to noise-aware methods**
>
> As we discussed in the general comment, we have now incorporated comparisons to noise-aware methods.
>
> At a high level, for the majority group, we find that group calibration remain resilient to low rates of label error (below 25 percent). At highr rates (>30 percent label error), we start to see declines in these performance metrics. However, for the minority group (30 percent of the dataset), we observe that the disparity metrics show consistent sensitivity to label error. This finding suggests that noise-aware methods show disparate performance in their ability to confer robustness to label error depending on data group sizes. A similar observation has also been made for other algorithmic interventions like Pruning (Tran et. al. 2022, “Pruning has disparate effect on model accuracy” & Hooker et. al. 2022 “What do deep neural networks forget”), Differential Privacy (Bagdasaryan et. al. 2018, “Differential privacy has disparate impact on model accuracy”), and Selective Classification (Jones et. al. 2021, “Selective Classification Can Magnify Disparities Across Groups”) and adversarial training (Xu et. al. 2021 “To be robust or to be fair: Towards fairness in adversarial training”).
>
> **Compare IF-Calib-Label to Noise-Aware Algorithms**
>
> Thank you for the suggestion. We have performed the comparison that you request. First, we note that several noise-aware algorithms do not incorporate a module to **explicitly** identify noisy labels in the training data. The output of these algorithms is an already trained model that performs well on held-out data despite having been trained on data that has label error. In this work, we are interested in an approach that explicitly identifies problematic examples, so that they can either be sent to a human to relabel or automatically relabeled. Second, our goal is  not to simply identify generic mislabeled examples, more specifically, we seek the particular mislabeled samples that have a high effect of the model's disparity metrics for a particular group. Most current noise aware algorithms are tailored to identifying mislabeled examples to improve validation/test set accuracy on the entire dataset.
>
> We have now included two noise-aware algorithms (MEIDTM, CVPR’22, and Confident Learning, Northcutt et. al. 2022, “Confident Learning: Estimating Uncertainty in Dataset Labels.”) in our baselines. In both cases, these approaches estimate a confidence/transition matrix on training samples to determine the probability of error of a sample's label for a given class. We compare against both approaches and find that they both underperform the influence-based approaches on the datasets considered. We updated the text (Appendix F) to discuss these results.
>
> As we previously noted, we caution that such a comparison might not be fair for these algorithms since they were designed for identifying and correcting generic label error instead of those that disproportionately affect a model's disparity metric.
>
>
> Thank you for the feedback, we hope we have adequately addressed your concerns. We will be happy to answer any additional questions. We encourage them to reconsider their score in light of our updates.

---

> > ### Author Response · Authors · 2022-12-01
> > **Happy to provide additional clarification**
> >
> > We hope our response clarified your initial concerns/questions. We would be happy to provide further clarifications where necessary.

---

> > > ### Author Response · Authors · 2022-12-06
> > > **Have your previous concerns been addressed?**
> > >
> > > We are reaching out to check whether the reviewer has any additional questions based on our previous response.

---

### Official Review · Reviewer_Xn1n · 2022-10-25

**Confidence:** 4
**Correctness:** 3
**Technical Novelty And Significance:** 2
**Empirical Novelty And Significance:** 1
**Recommendation:** 5

**Clarity, Quality, Novelty And Reproducibility:**

This paper generally is well-written and easy to follow, but most discussions are based on experimental results obtained from a few datasets. The experimental settings and comparison should be more detailed and comprehensive.

**Details Of Ethics Concerns:**

I have not found any ethics concerns.

**Strength And Weaknesses:**

Strength:
+ The research problems are important and may have many practical applications. The real-world machine learning dataset can easily contain label errors. Improving the robustness of learning models trained on noisy data is important. Existing methods mainly focus on downstream accuracy, but group-based disparity metrics have been ignored which are also important for designing a robust algorithm.
+ The proposed method is well-motivated. Estimating the influence of a single training input on a model’s group disparity metric is important for confident example selection and dataset purification.



Weakness:
+ The technical insight may not be enough. The authors have empirically illustrated that minority groups are more sensitive to label errors than majority groups. To make the conclusion more meaningful and practical, I think it would be great to add some theoretical analysis on the influence of label errors with different minority and majority group sizes.

+ The proposed method for estimating the ‘influence’ of perturbing a training point’s label on a disparity metric may not practical. The computational cost of the method seems very expensive and needs a lot of retraining processes to detect the effect of all training inputs, which can be hard to apply to a dataset with high-dimensional features. In addition, to demonstrate the performance of the proposed methods, some SOTA methods should be compared (e.g., JoCoR, CVPR’20; DivideMix, CVPR’20; MEIDTM, CVPR’22). The benchmark datasets such as CIFAR10 and CIFAR100 with different types of synthetic noise should also be compared.


+ The experiment setting is not clear to me. For example, it is not clear how the minority group and majority group in Fig. 1 and Fig.2 are obtained. I think the authors may also need to discuss that how to apply the convolutional network Resnet-18 to tabular and text datasets.


**Summary Of The Paper:**

This paper studies the effect of label error on the model’s disparity metrics (e.g., calibration, FPR, FNR) on both the training and test set. Empirically, the authors have found that label errors have a larger influence on minority groups than on majority groups. To mitigate the impact of label errors, The authors have proposed a method to estimate the influence of changing a single training input’s label on a model’s group disparity metric.

**Summary Of The Review:**

For me, the motivation and research problems of this paper are strong and important. My major concerns are that the technical contribution may not that strong, and the proposed method may not practical and hard to be applied to real-world machine learning datasets.

---

> ### Author Response · Authors · 2022-11-17
> **Part 1 of Response to reviewer Xn1n**
>
> We thank you for your comments and feedback. In addition to the general updates, we address your concerns here.
> ___
>
> >The technical insight may not be enough. The authors have empirically illustrated that minority groups are more sensitive to label errors than majority groups. To make the conclusion more meaningful and practical, I think it would be great to add some theoretical analysis on the influence of label errors with different minority and majority group sizes.
>
> As we discussed in the general comment, the three key contributions of this paper address challenges that were previously not understood in the label error literature. Specifically, the effects of label error (noise) on a model's average validation performance, empirically and theoretically, has been studied before. However, little was known, in the literature, about the effect of label error on a broad swath of model disparity metrics. In this paper, we address this challenge. In addition, we have now included a theoretical result that indicates that relabelling the problematic inputs that influence functions prioritizes reduces a model’s excess risk (under certain assumptions), and hence group calibration.
>
> >The proposed method for estimating the ‘influence’ of perturbing a training point’s label on a disparity metric may not practical.
>
> In recent work, Schioppa et. al. (2022)(Scaling up influence functions) scale influence functions up to large transformer models with about 300 million parameters. To do this, they use Arnoldi iteration to approximate the hessian with a diagonalized matrix that can be easily inverted. Following their approach, we have now switched our implementation to theirs and can confirm that it scales to Resnet-50 models. This means that our approach can be easily applicable to large scale models.
>
> > ... needs a lot of retraining processes to detect the effect of all training inputs, which can be hard to apply to a dataset with high-dimensional features.
>
> Our proposed approach does not need retraining. As a matter of fact, the influence functions approach was originally developed to help side step the need to retrain models in order to estimate the effect of a training sample. As indicated in Equations 4 and 5 from the text, given an already trained model, we simply need to perform an implicit hessian vector product for each training point. One can iterate through all training points to perform such a product to estimate each point's influence.
>
> **Comparison to noise-aware methods**\
> Thank you for sharing the references to these noise-aware methods. We have now incorporated these approaches into our settings. As we note in the contributions, we first identify problematic instances and then relabel these instances to improve the downstream model.
>
> The goal and experimental setting of most noise-aware algorithms differ from our setup in two ways: 1) We are interested in group disparity, so we are focused on settings and datasets that includes a protected group attribute (often Race, Gender, and others in the fairness literature), and 2) We primarily seek to identify the problematic instances that need to be relabelled (often by a human labeler), and not automatically learn a model that is robust to label error. The output of our proposed approach is a set of points that should be relabelled by a human, whereas in noise-aware learning, the goal and output is a new model that is robust to label noise. Cifar-10 and Cifar-100 do not have protected attribute labels, so we did not originally experiment on these datasets.
>
> We have now performed additional experiments using the modified Cifar-10 dataset from Hall et. al. (2022), "A Systematic Study of Bias Amplification". In this paper, the authors modify Cifar-10 to be a binary classification task, and then inject group labels into the two classes by inverting a fraction of the examples in each class. Given a specified parameter $\epsilon \in [0, 1/2]$, a $\frac{1}{2} - \epsilon$ of the negative class is inverted, while a $\frac{1}{2} + \epsilon$ of the positive class is inverted leading to $2\epsilon$ fraction of one group of samples and $1-2\epsilon$ of the other group. In all experiments we set $\epsilon=0.15$ for a 30 percent minority group membership.
>
> We replicate our experimental setup on this task where we inject label noise into the training set. We test the MEIDTM, DivideMix, and a robust loss approach. We find consistent results across these approaches (New Figure 4 in text). At a high level, for the majority group, we find that model accuracy and downstream disparity metrics remain resilient to low rates of label error. At higher rates, we start to see declines in these performance metrics. However, for the minority group, the disparity metrics show consistent high sensitivity with increased label error. This finding suggests that noise-aware methods show disparate performance in their ability to confer robustness to label error depending on data group sizes.

---

> > ### Author Response · Authors · 2022-11-17
> > **Part 2 of Response to reviewer Xn1n**
> >
> >
> > #### Comparison to noise-aware methods (Continued)
> >
> > The aforementioned finding suggests that noise-aware methods show disparate performance in their ability to confer robustness to label error depending on data group sizes. A similar observation has also been made for other algorithmic interventions like Pruning (Tran et. al. 2022, “Pruning has disparate effect on model accuracy” & Hooker et. al. 2022 “What do deep neural networks forget”), Differential Privacy (Bagdasaryan et. al. 2018, “Differential privacy has disparate impact on model accuracy”), and Selective Classification (Jones et. al. 2021, “Selective Classification Can Magnify Disparities Across Groups”) and adversarial training (Xu et. al. 2021 “To be robust or to be fair: Towards fairness in adversarial training”).
> >
> > Taken together, the results above suggest that our proposed identification and relabelling strategy addresses shortcomings of current noise-aware approaches. We plan to perform a more comprehensive empirical assessment for the final version of the paper.
> >
> > **Clarification of Experimental Setting**
> > 1. **Majority Group and Minority Groups:** For each dataset, we have access to feature annotations that partition the dataset into groups. As indicated in Table 1, these annotations are group variables like Sex (Male, Female), and Race. For each group annotation, we can partition the training set into groups; the training subset with the largest size is the majority group, while the training subset with the smallest size is the minority group. We've updated the text to reflect this.
> > 2. **Tabular Data:** To train CNNs on tabular data, we first apply a simple 2-layer MLP on the tabular dataset. We then simply extract representations from the hidden layer of the MLP, reshape these to be (32 by 32) two-dimensional and normalized ([0-1]). This strategy has been previously employed to train deep learning models on tabular data previously (Borisov et. al. 2022, “Deep Neural Networks and Tabular Data: A Survey”). We've updated the text to clarify this as well.
> >
> > Thank you for the feedback, we hope we have adequately addressed your concerns. We will be happy to answer any additional questions. We encourage you to reconsider your score in light of our updates.

---

> > > ### Author Response · Authors · 2022-12-01
> > > **Happy to provide additional clarification**
> > >
> > > We hope our response clarified your initial concerns/questions. We would be happy to provide further clarifications where necessary.

---

> > > > ### Comment · Reviewer_Xn1n · 2022-12-02
> > > > **My concerns has been partially solved**
> > > >
> > > > Dear Authors,
> > > >
> > > > Thank you very much for the response. My concern about the computational cost has been addressed. However, I still have concerns about the technical novelty.
> > > >
> > > > + The technical results in this paper seem highly dependent on the previous paper "Understanding black-box predictions via influence functions". Specifically, From Eq (1) to Eq. (3), all the results are proposed by the previous paper.
> > > >
> > > > + Theorem 1 is not very interesting to me. It basically says that by improving the data quality, the expected calibration error will become smaller, which is trivial.
> > > >
> > > > It would be great if a more concrete clarification of the technical contribution of this paper can be provided.

---

> > > > > ### Author Response · Authors · 2022-12-02
> > > > > **Addressing Technical Novelty**
> > > > >
> > > > > Thank you for your feedback, and challenging us to more clearly delineate our technical contributions. We provide additional clarification here.
> > > > >
> > > > > ___
> > > > >
> > > > >
> > > > > **Difference from the Koh et. al. (Understanding black-box predictions via influence functions)**
> > > > > >The technical results in this paper seem highly dependent on the previous paper "Understanding black-box predictions via influence functions". Specifically, From Eq (1) to Eq. (3), all the results are proposed by the previous paper.
> > > > >
> > > > > **Answer**: The reviewer is correct that Eqns 1-3 are due to Koh et. al. We do not claim these as contributions; we provide these to aid the reader. Our key contributions in that section are Eqns 4-5. We now provide additional clarification.
> > > > >
> > > > > In using influence functions, our goal is to: 1) identify training samples whose *label have a high influence on any disparity metric of interest (e.g. group calibration, false positive rate, false negative rate, etc.)*, and 2) propose a method to improve these disparity metrics via relabeling.
> > > > >
> > > > > In Koh et. al., they propose the influence functions to rank training samples that have a high influence on 1) the parameters, and 2) a single test example's loss or prediction. We extend these results to fairness metrics. To reinforce our point that such an extension is needed, the results of Fig. 5 demonstrate that a direct application of the original influence functions approach does not capture group-based effects that is often key for improving fairness metrics.
> > > > >
> > > > > Eqns 4 and 5 directly address these challenges, and allow us to repurpose influence functions for addressing *fairness* challenges. Our presentation in Section 4 is sparse, due to space constraints.  However, we will add a paragraph to more clearly discuss these key differences. We will also expand the derivation in the Appendix to more clearly show these differences as well.
> > > > >
> > > > >
> > > > > **A note about the relabel-and-finetune scheme**\
> > > > > We saw a previous comment about the relabeling portion of the paper. To clarify, we have now compared to schemes like DivideMix and MEIDTM. We discuss these results in Sections 3.3 (Noise-aware robust learning has disparate impact) and 5.1(Identifying label error) in the updated draft.
> > > > >
> > > > > A point of caution first, since our setting requires sensitive group variables, we cannot directly use Cifar-10 and Cifar-100. We modify Cifar-10, as we discussed in the part 1 of the previous comment, to inject group annotations. In these experiments we make two important findings:
> > > > >
> > > > > 1. Previous relabeling methods provide improvements for the majority group in the data, but not the minority group;
> > > > > 2. Our proposed group-based relabelling scheme outperforms these approaches since it directly identifies minority samples whose label have a high effect on that group's disparity metric.
> > > > >
> > > > > These two findings demonstrate that our proposed scheme provides benefits that current approaches donot address.
> > > > >
> > > > > **Theorem 1 is trivial**\
> > > > > We respectfully disagree with this opinion; the insight of theorem is obvious---in hindsight. However, we agree with the reviewer that the theorem's takeaway that if one relabels mislabeled examples, on a group basis, it leads to an improvement in group calibration is not surprising.
> > > > >
> > > > > In a previous result, Kong et al. show that if one relabels by influence, the *average* validation loss, across all samples, of the fine-tuned model is reduced. However, as we discussed earlier, prioritization by generic influence favors the majority group, so it is not obvious that relabeling should improve **group calibration**, even for the minority group. In addition, the relationship between group calibration and the average validation loss is unclear. The key insight here is a way to relate a per-group average loss to that group's calibration. We are currently unaware of any previous result that demonstrates this point. Even though the theorem's result is not surprising, it does provide justification for the scheme we proposed.
> > > > >
> > > > > We thank the reviewer again for the feedback, and we will be happy to answer any additional questions that you have.

---

> > > > > > ### Author Response · Authors · 2022-12-06
> > > > > > **Have your concerns around technical novelty been addressed?**
> > > > > >
> > > > > > Hello,
> > > > > >
> > > > > > We wanted to reach out again to ask whether you have any additional concerns regarding technical novelty.

---

### Author Response · Authors · 2022-11-17
**General Response and Summary of Updates to Manuscript**

We thank the reviewers for noting that we address an important problem (Xn1n, oCK7), with an interesting analysis (DK6U) that is very well organized, written, and easy to follow (DK6U, oCK7). First, we provide a high-level summary of the changes that we've made to the draft to address your feedback, and conclude with an overview of our key contributions, and how they differ from previous work.
___

Here is the summary of updates that we've made to the draft:

- Added a new section discussing empirical results on the sensitivity of models trained using noise-aware algorithms. The previous disparate effect that we observed for minority groups persists even for models trained with noise-aware algorithms (**Reviewers Xn1n and oCK7**).
- Revamped related work section to clarify our findings and differentiate our contributions from previous work. We have also incorporated all the references provided by the reviewers (**Reviewers Xn1n, oCK7, & DK6U**).
- Added empirical results on a new image and a text dataset and find that our results remain consistent across these datasets (**Reviewers DK6U**).
- Added new noise-aware baselines to label error identification experiments, and show that our proposed approach outperforms these approaches (**Reviewers Xn1n and oCK7**).
- Finally, we now provide a theoretical guarantee for the automatic fix proposed in the work. This theorem suggests that our proposed relabeling leads to models with provably improved group calibration (**Reviewers Xn1n and oCK7**).

To end this update, we discuss a common concern across all reviewers.

**Novelty and Technical insight**\
The impact of label error on model accuracy is a well-studied problem. However, the effect of label error on a model's disparity metrics is still poorly understood in the literature. Performance metrics estimated on the entire validation (or test) set are typically invalid for more fine-grained groups in the dataset (See: Distributionally Robust Losses for Latent Covariate Mixtures, Duchi et. al. 2022). Consequently, we cannot expect results on the effect of label error on validation loss (or other performance metrics) to provide insights on how label error affects group calibration and other disparity metrics. Consequently, in this paper we characterize the impact of label error on these disparity metrics.

To contextualize the results in this paper, we now summarize our key contributions:

1. **Empirical demonstration of sensitivity of a model's disparity metrics to label error:** First, we find that a model's group-based disparity metrics are sensitive to the presence of label error in either the training and/or test data. In particular, we find that such sensitivity is more pronounced for samples in the minority group in the data. We have now conducted additional experiments that also show that such sensitivity remains unaffected, for minority groups, even when models are trained with noise-aware algorithms. Consequently, our findings suggest that the presence of label error can render the results of a fairness audit unreliable.
2. **Approach to identify training inputs whose labels have high effect on any differentiable disparity metric:** Having established that a model's disparity metric is sensitive to label error, we then provide a way to identify the training samples whose wrong labels have the most influence on any differentiable disparity metric of interest. Our proposed formulation departs from previous approaches in a few ways. First, as we show in the results of Figure 4, simply prioritizing samples based on standard versions of training sample influence often identifies training samples in the majority class. Hence, the effect of label error on the minority group, the group most disproportionately affected, is often ignored with naive influence-based ranking. Second, we consider a more fine grained notion of influence where we characterize the impact of the change in label on the disparity metric of interest. Both of these specializations enable improved performance as the ablation on Figure 4 shows, and depart from the way influence functions were previously used in the literature.
3. **Correcting Label Error:** Lastly, in Section 5.2, we present an automatic relabel-and-finetune scheme that produces updated models with improved group calibration error. We compare this strategy to several others (now including noise-aware baselines), and find that our approach outperforms these other strategies. In addition, we now provide a theorem that demonstrates that the proposed strategy provably improves group calibration.

Taken together, these contributions represent novel insights to the literature on the effect of label error on a model’s disparity metrics.

---

### Decision · Program_Chairs · 2023-01-20

**Decision:**

Accept: poster

**Justification For Why Not Higher Score:**

Reviewers agree that the studied problem is important and may have many practical implications and that the proposed method is well-motivated.

**Justification For Why Not Lower Score:**

Reviewers also have several sensible concerns; e.g., the technical contribution may not be strong enough, and the proposed method may not practical to deal with real-world machine learning datasets.

**Metareview: Summary, Strengths And Weaknesses:**

This paper investigates the effect of label error on the model’s disparity metrics (e.g., calibration, FPR, FNR) on both the training and test set. The authors found that empirically, label errors have a larger influence on minority groups than on majority groups. The authors proposed a method to estimate the influence of changing a single training input’s label on a model’s group disparity metric. Reviewers agree that the studied problem is important and may have many practical implications and that the proposed method is well-motivated. At the same time, reviewers also have several sensible concerns; e.g., the technical contribution may not be strong enough, and the proposed method may not practical to deal with real-world machine learning datasets. However, overall, I believe the value overweights the issues in the paper.

**Note From Pc:**

if the above contains the word "oral" or "spotlight" please see: "oral" presentation means -> notable-top-5% and "spotlight" means -> notable-top-25%. As stated in our emails, we are disassociating presentation type from AC recommendations